# Occupational Risk of Airborne *Mycobacterium tuberculosis* Exposure: A Situational Analysis in a Three-Tier Public Healthcare System in South Africa

**DOI:** 10.3390/ijerph181910130

**Published:** 2021-09-27

**Authors:** Dikeledi O. Matuka, Thabang Duba, Zethembiso Ngcobo, Felix Made, Lufuno Muleba, Tebogo Nthoke, Tanusha S. Singh

**Affiliations:** 1National Institute for Occupational Health (NIOH), National Health Laboratory Service (NHLS), Johannesburg 2000, South Africa; dikeledim@nioh.ac.za (D.O.M.); thabangd@nioh.ac.za (T.D.); ZethembisoN@nioh.ac.za (Z.N.); felixm@nioh.ac.za (F.M.); lufunom@nioh.ac.za (L.M.); TebogoN@nioh.ac.za (T.N.); 2Department of Clinical Microbiology and Infectious Disease, School of Pathology, University of the Witwatersrand, Johannesburg 2000, South Africa; 3Department of Environmental Health, School of Health Sciences, University of Johannesburg, Johannesburg 2028, South Africa

**Keywords:** TB, airborne transmission, occupational health, healthcare workers, environmental sampling, real-time PCR, TB infection control, ventilation, UVGI, hierarchy of controls

## Abstract

This study aimed to detect airborne *Mycobacterium tuberculosis* (MTB) at nine public health facilities in three provinces of South Africa and determine possible risk factors that may contribute to airborne transmission. Personal samples (*n* = 264) and stationary samples (*n* = 327) were collected from perceived high-risk areas in district, primary health clinics (PHCs) and TB facilities. Quantitative real-time (RT) polymerase chain reaction (PCR) was used for TB analysis. Walkabout observations and work practices through the infection prevention and control (IPC) questionnaire were documented. Statistical analysis was carried out using Stata version 15.2 software. Airborne MTB was detected in 2.2% of samples (13/572), and 97.8% were negative. District hospitals and Western Cape province had the most TB-positive samples and identified risk areas included medical wards, casualty, and TB wards. MTB-positive samples were not detected in PHCs and during the summer season. All facilities reported training healthcare workers (HCWs) on TB IPC. The risk factors for airborne MTB included province, type of facility, area or section, season, lack of UVGI, and ineffective ventilation. Environmental monitoring, PCR, IPC questionnaire, and walkabout observations can estimate the risk of TB transmission in various settings. These findings can be used to inform management and staff to improve the TB IPC programmes.

## 1. Introduction

Hospital-acquired tuberculosis (TB) infection caused by *Mycobacterium tuberculosis* (MTB) is an occupational health risk affecting healthcare workers (HCWs) in low- and middle-income countries [1,2,3,4,5,6,7,8,9] and is among the top ten leading causes of death globally [10,11]. Transmission of MTB occurs through the airborne route from aerosol droplets generated from exhaling, talking, sneezing, or coughing from an infected person [1,2,11,12,13]. Undiagnosed and/or untreated patients, [6,14,15] in addition to inadequate infection control (IC) measures often serve as the primary source of exposure to MTB [5,6,9,14,15,16,17].

According to the WHO Global TB report 2020, South Africa (SA) is among the top eight countries with the highest TB burden recorded at 615/100,000 population [11]. A meta-analysis conducted in 2017 reported a pooled estimated TB incidence rate of 97/100,000 among HCWs [18]. When compared with the general population, HCWs are three times more likely to be at risk of TB disease (25–5361/100,000 per annum) [1,7,18,19,20,21] with an estimated 81% of TB cases among HCWs in high-incidence countries [4] likely due to prolonged occupational exposure in health facilities [1,2,4,9,15,16,18,20,22,23]. The latter and the high prevalence of multidrug-resistant (MDR) and extensively drug-resistant (XDR) TB, further exacerbate the high TB mortality rate [11,20,23,24,25,26,27,28,29,30,31]. South African surveys have reported up to 16% of HCWs living with HIV, placing them at greater risk of developing TB in the workplace than HCWs without HIV [20,32,33,34,35,36,37]. The likelihood of TB transmission amongst health workers is also influenced by the patient population, type of occupation, regional TB prevalence, medical facility access [4,38], inadequate infrastructure, poor ventilation [39,40], delayed and incorrect diagnosis [41], and TB infection control programmes’ efficacy [4,38,40]. The resource allocation shift to the COVID-19 pandemic also poses a serious threat to the reduction of the global TB detection and notifications, thereby hampering the progress made in recent years [42,43,44]. Between January and June 2020, SA was among the four countries that reported a significant decrease in the number of TB-diagnosed people that account for 44% of global TB cases [11].

Despite the existence of the infection prevention and control (IPC) policies and guidelines in most healthcare settings, IPC remains poorly implemented in many healthcare facilities [45], especially in a high-TB-burden country such as South Africa as human behaviour and non-compliance remain a challenge [8,20,46]. Therefore, identifying the risk areas and barriers of IPC compliance would aid in prioritising and enhancing infection control (IC) interventions [45]. Previous studies have largely focussed on the level of exposure in settings where there are known active cases of TB [31,47,48] and on TB preventive treatment [11,49].

Only a few studies have characterised airborne MTB concentration profiles in health facilities [3,5,31]. A combination strategy of air sampling filtration and a sensitive and short turn-around-time molecular technique such as polymerase chain reaction (PCR) has been used in previous studies to detect airborne MTB [3,5,24,31,39,47,50,51]. This study aimed to investigate the levels of MTB in perceived risky areas of various types of healthcare facilities (HCFs) in three provinces of SA and to determine potential risk factors attributing to airborne MTB concentrations.

## 2. Materials and Methods

### 2.1. Study Design and Population

This was a cross-sectional study conducted at nine public sector HCFs in SA across three provinces (Gauteng, KwaZulu-Natal, and Western Cape). The provinces were grouped into a low-burden TB region, which was Gauteng [52], and high-burden TB regions, which were the Western Cape and the KwaZulu-Natal [17], based on reported TB prevalence in these provinces. A total of three district hospitals, four TB hospitals, and two primary health clinics (PHCs) were selected intentionally from these provinces to represent three types of HCFs. Samples were collected at perceived risk areas, including outpatient departments (OPD), TB wards, X-rays, triage, general medical wards, outpatient consulting rooms, occupational therapy (OT), casualty (emergency), waiting areas, and administration (control).

### 2.2. Facility Assessment

A walkabout was conducted before sampling by a team from the National Institute for Occupational Health (NIOH), a division of the National Health Laboratory Service (NHLS), which included occupational hygienists and medical scientists to identify possible TB risk areas within the hospitals for sampling strategy. Control measures such as administrative (IPC policies, training, and risk assessment records), patient management (identification and segregation of coughing patients, and education and awareness), engineering controls (natural and mechanical ventilation, isolation rooms, air disinfection devices), and personal protective equipment (particulate respirators) were also observed for each facility. A self-reported IPC questionnaire based on the approach from the 2009 WHO report [53] was completed by designated healthcare personnel at each facility, which gathered information on administrative, environmental, clinical, and occupational health measures.

### 2.3. Environmental Air Sampling

In total, 591 air samples (repeated for 2–3 days depending on shift) were collected across the nine facilities from March 2017 to October 2017. The number of air samples collected from each facility varied depending on the number of high-TB-risk areas and the size of the facility. Personal samples and stationary samples were collected by filtering air through a 37 mm polytetrafluoroethylene (PTFE) filter cassette at a flow rate of 4 L/min (*n* = 264) and 20 L/min (*n* = 327) using low- and high-volume Gillian sampling pumps (SKC Inc., Eighty Four, PA, USA), respectively. Stationary samples were collected at a height of 1.5 m to represent the average standing height of a health worker’s respirable zone. Health workers wore the sampling cassette on the collar lapel in their breathing zone for approximately 470 min (±7.85 h). Sampling pumps were calibrated before and after sampling using a Gilibrator bubble flow meter (Sensidyne, St. Petersburg, Florida, FL, USA), and a difference in flow rate of less than 5% was accepted [54]. Field blanks were also collected in each facility to exclude sources of contamination by exposing the filters to the same field conditions as the samples. The samples were transported at ambient temperature and stored at −20 °C prior to analysis. An IAQ-Calc TSI model (TSI Instruments Ltd., High Wycombe, UK) was used to measure temperature and relative humidity. The TSI VelociCalc model: 9555-p (TSI Instruments Ltd.) was used to measure airflow per area and the TSI Accubalance Balometer Model 8371, S/N 97030449 (Thermo-System Instruments, Shoreview, MN, USA) was used to measure the airflow from diffusers and grilles or the airflow entering exhaust outlets (supply and return air on the ventilation system), repeated for 2–3 days depending on shift. Air changes per hour (ACH) were calculated using ventilation measurements and room dimensions. UV irradiance was measured 1 and 2 m away from the fixture and at occupant’s eye levels using a Goldilux UV–C meter, S/N 823749 (Measuring Instrument Technology, Pretoria, South Africa) in facilities that had UVGI fixtures.

### 2.4. Detection of Airborne M. tuberculosis Using Real-Time Quantitative Polymerase Chain Reaction

#### 2.4.1. Preparation of the *M. tuberculosis* DNA for the Real-Time Quantitative Polymerase Chain Reaction

*Mycobacterium tuberculosis* H37Ra (ATCC 25177) cells grown aerobically in MGIT tubes for 21 days were vortexed and harvested. The optical density (OD) of 0.5, which is approximately 10^8^ cells/mL of MTB, was measured using a densitometer. The cells were serially diluted (10-fold) to 10^1^ cells/mL. The *M. tuberculosis* H37Ra (ATCC 25177) was used to construct a standard curve (LightCycler^®^ software version 4.1, Roche Diagnostics GmbH, Mannheim, Germany).

#### 2.4.2. DNA Extraction

The *M. tuberculosis* DNA was aseptically extracted from the PTFE filters in a 2 mL stripping solution (containing 1% Triton X-100 in 10 mM Tris-HCl, pH 8.0) by shaking for 45 min. The DNA was stored at 4 °C until further use. A sterile filter was included in the test as a lab negative control.

#### 2.4.3. Quantification of *M. tuberculosis* Using Real-Time Quantitative Polymerase Chain Reaction

Quantification of *M. tuberculosis* isolated from the filters was done using real-time quantitative polymerase chain reaction (qPCR) using the LightCycler 1.5 instrument (Roche Diagnostics International, Rotkreuz, Switzerland). The LightCycler *Mycobacterium* Detection Kit (Roche products (Pty) Ltd., Randburg, South Africa) targeting the 16 S ribosomal RNA (rRNA) gene was used according to the manufacturer’s instructions. The kit is based on real-time PCR technology for detecting *M. tuberculosis*, *M. avium,* and *M. kansasii* by amplifying the 16 S rRNA gene including the hypervariable region A at a channel of 640. Samples and standards were amplified in triplicate. A melting curve analysis at 55.5 °C was performed to determine the melting temperature (Tm) values for the sequences targeted by the hybridisation probes. The Tm values were automatically assigned from a plot generated by the instrument. A Tm of 53.5–56.5 °C indicated *M. tuberculosis*. Unknown DNA concentrations expressed as DNA copies/µL were extrapolated using linear regression from the standard curve. Results were accepted if the controls passed and the efficiency of the standard curved used to extrapolate the concentrations of the unknown samples was ≤2 as recommended by the kit manufacturer (Roche, Germany) and the coefficient of variation (CV) between replicates was ≤20%. The LightCycler *Mycobacterium* detection kit has a limit of detection (LOD) of 0.028 target copies/mL with a 95% CI. For quality control, field blanks (5%), laboratory positive and negative controls, and kit positive and negative controls were included in the analysis. The final MTB airborne concentration (DNA copies/m^3^) was calculated using the number of MTB DNA copies/mL, sampling time, and flow rate.

### 2.5. Statistical Analysis

Statistical analysis and data cleaning were carried out using Stata SE version 15.1 software (STATA, 14905 Lakeway drive, College Station, TX, USA). Data were checked for missing values and outliers. Data were described in numbers and percentages for categorical variables, while continuous variables were summarised as median and interquartile ranges. Comparison of continuous variables by categorical variables were done using two-sample independent *t*-test, while Wilcoxon rank sum tests and Kruskal–Wallis tests were used to assess median difference between measured parameters and factor variables. The association factors of ultraviolet germicidal irradiation (UVGI) and airborne *M. tuberculosis* over time were assessed using generalised linear mixed-effect logistic regression modelling with random intercept considered at the individual level (samples). The analyses were adjusted for other covariates. The regression estimates were reported in odds ratios (ORs) and 95% confidence intervals (CIs).

## 3. Results

In this study, air filtration and PCR methods were used to measure MTB concentrations in different areas of nine HCFs (four TB hospitals, three district hospitals, and two primary healthcare clinics (PHCs)). A self-administered questionnaire gathered information on IPC and occupational health measures and the reported findings are described in Table 1.

Administrative controls: Two TB hospitals and one district hospital reported not having a dedicated nurse to treat TB patients. One TB hospital and one district hospital did not conduct a TB risk assessment. All TB and district hospitals (100%) had an IPC policy and HCWs were trained on this policy, while one PHC had no IPC policy. However, all nine facilities reported the training of HCWs on TB IPC. A TB register was available only in three TB hospitals and one district hospital. Half of both TB hospitals and PHCs as well as one district hospital did not report the bed capacity, although all nine facilities reported the total number of staff.

Environmental control measures: All PHCs, three TB hospitals, and one district hospital had a dedicated sputum production area. All TB hospitals and PHCs used natural ventilation through open windows on the opposite sides of the walls, with TB hospitals’ windows opening directly to the outside. District hospitals (100%) and TB hospitals (75%) used mechanical ventilation. In addition, 3 of 4 TB hospitals and a district hospital used UVGI for air disinfection, which was functional. However, high-efficiency particulate air (HEPA) filtration and negative pressure were only used by two district hospitals.

Clinical control measures: All these measures (patient screening on arrival, designated area for screening, separate rooms for TB patients, and TB awareness material) were implemented in all PHCs but not consistently practiced in district and TB hospitals. All nine hospitals had TB IPC education programmes for patients; however, material was not available in some facilities (25% for TB hospitals and 66.7% for district hospitals during the visit.

Occupational health (OH) control measures: All TB hospitals reported that they have a respiratory protective programme, and all staff were screened for TB and did baseline chest X-rays. All nine facilities reported the risk of contracting TB and used N95 respirators, but only one district hospital lacked access to occupational health services. None of the HCWs in the district and the PHCs had undergone respirator fit testing. Several facilities (78%) reported cases of TB among HCWs, with TB hospitals (20 cases) leading, followed by the district hospitals (10 cases) and PHCs (7 cases).

A total of 591 airborne *Mycobacterium tuberculosis* (MTB) samples were tested, of which, 578 (97.8%) were negative and 13 (2.2%) were positive (Table 2). Most positive samples (3.2%) were from the Western Cape province. District hospitals had the highest percentage of positive samples (5.17%), while TB hospitals were mostly negative (98.8%) and four (1.2%) were positive. No MTB-positive samples were detected in primary health clinics. Airborne MTB was higher when UVGI was absent (*n* = 9) than when present (*n* = 4). The distribution of the number of positive samples was similar between natural and mechanical ventilation. All average environmental parameters (temperature, relative humidity, carbon dioxide) except velocity were slightly higher in areas where positive samples were detected compared to areas without airborne MTB. Air changes per hour ranged from 0 to 0.34 ACH for positive sample areas and 0 ACH for negative sample areas.

Airborne MTB was detected in district and TB-specialised hospitals of the Western Cape and Gauteng provinces, with DNA copies ranging from 1.08 × 10^4^ to 3.55 × 10^7^ DNA copies/m^3^ and included all area samples, no positive MTB was detected in personal samples. The MTB-positive areas included medical wards, casualty, and TB wards (Table 3). No airborne MTB was detected in the other areas sampled.

A summary of environmental parameters according to province, hospital type, season, UVGI, and ventilation is described in Table 4. There was enough evidence to suggest that the median values of humidity and velocity were significantly higher in the Western Cape province than in Gauteng province. There was no difference in temperature and carbon dioxide between these two provinces. The primary healthcare clinics (PHCs) had higher temperature and humidity measurements than the other two facilities (*p* = 0.0001). The distribution for temperature, humidity, and velocity differed significantly between the summer and winter sampling periods. Temperature, humidity, and carbon dioxide had significantly higher median values when UVGI was absent, unlike velocity. There were significantly higher temperature median differences for mechanical ventilation than in natural ventilation. Humidity was higher in naturally ventilated than in mechanically ventilated areas.

Factors associated with airborne *Mycobacterium tuberculosis* are shown in Table 5. Increased number of days increased the probability of detecting TB, while the installation of UVGI decreased the odds of detecting MTB, all without statistically significant difference. The Western Cape province was 4.82 times more likely to be positive for airborne MTB DNA compared to Gauteng province with statistically significant difference (OR: 4.82 95% CI: 1.091–21.358). The odds of detecting airborne MTB were significantly higher with mechanical ventilation than with natural ventilation (OR: 4.77; 95% CI: 1.396–16.280). Every unit increase in carbon dioxide statistically significantly increased the odds of TB by 0.3% (95% CI: 1.0001–1.0046).

## 4. Discussion

All nine facilities had trained staff on TB infection prevention and control (TBIPC) and implemented TB educational programmes for patients and most facilities had conducted risk assessment except for two (TB and PHCs) in this study. However, this is contradictory to other SA studies where a lack of TBIPC training of health workers was reported [16,19] and some PHCs provided in-service infection control training [17,46]. The discrepancy may be due to a small hospital sample size in this study. TBIPC is an important component in overall TB control efforts within health facilities [9,42]. Our study demonstrated that the use of natural ventilation was variable between the different facilities. Adequate ventilation through opening of windows (in addition to respiratory protection) resulted in low concentrations of airborne MTB [39] and is among effective infection control measures in healthcare [55].

Occupational health controls (TB screening of staff, respiratory protection programme with fit testing, baseline X-rays, OH services) were fully functional for the TB hospitals but was, however, poor for district hospitals (Table 1), probably due to the awareness of the possible high risk of exposure due to diagnosed referral patients. District hospitals reported patient screening (66.7%) and separation of TB patients; however, the open-window policy (natural ventilation) was lacking and the TB register, educational material, and a dedicated sputum production room for TB patients were poorly implemented, and these may be the contributing factors to the highest percentage of positive samples in these facilities. The lack of TB screening policies for HCWs also hampers effective IPC implementation [5,20]. Although workers in all facilities were provided with N95 respirators, fit testing was, however, lacking in district hospitals and PH clinics, and this may expose HCWs in the workplace if respirators do not fit properly. The correct use and sufficient supply of respirators are essential measures for preventing TB transmission in HCWs [7,16].

This study demonstrated variability in airborne exposure between provinces, facility type, and season. The Western Cape province had the highest positive MTB samples, followed by Gauteng, and no MTB was detected in KwaZulu-Natal in this study. The detectable levels of TB DNA were higher in district hospitals (3.63%) than in TB hospitals (1.58%) despite them reporting patient screening (100%). The rate of positive samples (2.2%) in the current study was less than those reported in our previous TB pilot study (8.3%) [5], Thailand (3%) [31], Slovenia (44.4%) [39], and Taiwan (63.8%) [3]. All 13 positive samples were detected in the winter season, possibly due to windows being closed during the cold weather. The negative TB air samples in the summer season may be a consequence of the windows being kept open at the time of sampling. Amongst other factors, environmental (physical) parameters and air cleaning or disinfection can impact on the airborne transmission of MTB [5] by causing variability in results that was significantly different between summer and winter seasons in this study.

In the current study, higher airborne positive MTB was detected in casualty (emergency department), followed by TB wards and medical wards. In comparison, the levels were higher in waiting rooms and consulting rooms of medical departments, followed by emergency and medical wards and significantly higher in TB areas than in non-TB areas [3]. The higher airborne MTB levels in the emergency department (casualty) may be due to patients being at early stage of TB disease without treatment [3]. In contrast to our results, airborne droplets of MTB have not been reported in emergency departments before [31] but only in medical wards where TB patients are treated [3,48]. The latter is supported by findings where the majority of HCWs diagnosed with TB did not work in TB wards or TB clinics and, thus, may have contracted TB from close contact with patients [41] as most TB patients usually first start at general medical departments where doctors and nurses are the primary caregivers [2,28,36]. Other contributing factors include geographical factors, different settings, population TB incidence, detection kit sensitivity, strict implementation and compliance of IPC measures in various states [31], ventilation design, patient condition [3], as well as patients loads and waiting times [46]. Furthermore, factors such as poor infrastructure, staff shortage, and financial constraints may also have contributed to positive airborne MTB, as reported by some facilities during interviews in our study, which is similar to other reports [8,9]. However, another study reported extensive exposure to TB patients in medical wards with known TB patients and in casualty (emergency department) where contact with suspected patients is possible [16]. Higher incidence of TB was also reported in HCWs working in TB wards than those with no work history in TB ward [20]. Airborne particles ≤ 3 µm (most airborne bacteria and viruses) were found to travel 9.5 m within 5 min from a general patient room to a nursing station and 14.5 m in 14 min to an isolation anteroom entrance at concentrations 2–5 times higher than the ambient levels [56].

Airborne MTB was not detected in primary health clinics (PHCs) during the time of sampling, and this may be due to the early screening of patients on arrival, number of patients, and proportion of infectious patients and severity of disease during sampling, sufficient natural ventilation and dilution effect, and comprehensive clinical control practices implemented. This contradicts reports of poor TB-IC implementation in PHCs [8,17,46], whereas the majority of negative samples in TB hospitals may be attributed to several factors including use of natural ventilation through open windows (100%) with unrestricted flow, dedicated sputum production area, UVGI devices for air cleaning, IPC policy, effective OH services, and training of staff on TB control practices. Open windows enable adequate airflow and dilute airborne particles [39,46,57]. Two-thirds of facility types with designated areas for sputum production display good practice, which will aid in limiting the spread and transmission of the disease.

Most positive samples (4.17%) being detected where mechanical ventilation existed in this study may be due to the malfunctioning of these engineering controls, while natural ventilation showed reduced airborne MTB DNA (98.4% negative samples). Although a few facilities reported the use of mechanical air cleaning methods, some lacked maintenance, which may give HWs a false sense of security, especially if they were not functioning optimally. All district hospitals (100%) had mechanical ventilation, whereas only one had HEPA filtration and another had negative pressure (Table 1).

The ASHRAE recommended ventilation rate of 6 ACH in patient rooms of hospitals [31,58] was not met in all sampled areas (0.0–0.34 ACH) in this study (Table 2). In another study, the ventilation system was capable of removing almost 60% of airborne particles in a patient room. In addition, negative pressure and door position prevented the aerosols from the patient ward from reaching the isolation rooms [56]. In contrast, airborne MTB was detected in the negatively pressured isolation rooms with TB patients although lower than in a medical ward with a suspected TB patient who was diagnosed with TB a month later [3]. Hospital settings pose a serious risk of tuberculosis (TB) transmission to HCWs [20,39]; therefore, the presence of airborne TB bacilli probes the need to initiate additional mitigation strategies for MTB transmission.

Four air cleaning methods (natural and mechanical ventilation, UVGI, HEPA, and negative pressure) were reported to be used amongst the nine facilities. The presence of UVGI devices in MTB-negative areas may have contributed to the lower detection of the DNA in comparison to positive areas where ultraviolet germicidal irradiation (UVGI) was absent. The upper room UVGI showed approximately 86% efficacy in a study conducted in Witbank, South Africa [5,59]. In a controlled laboratory setting, 46% (6 of 13) of UVGI devices tested showed 100% effectiveness in killing aerosolised MTB, while the remaining devices’ effectiveness ranged from 43.7% to 95.1% [60]. A UVGI air disinfection device (total fixture output (of 15–20 mW/m^3^ or whole-room UV irradiance of 5–7 μW/cm^2^) coupled with air mixing was highly effective in reducing tuberculosis transmission under hospital conditions [61]. Walkthrough observations revealed that the UVGI fixtures installed were not serviced, and no records were available at some hospitals during the survey. This is similar to previous findings reporting non-functional UVGI and the absence of maintenance records for the past eight years [5]. The findings from this study have confirmed that the presence of UVGI and natural ventilation may be effective in reducing airborne MTB, contrary to the absence of UVGI in areas with positive MTB samples. Infection preventative control measures according to the hierarchy of controls have been proven to reduce transmission of TB in hospitals [7].

Seven of nine hospitals (77.7%) reported TB cases in their respective workplaces (Table 1), confirming that occupational TB still poses a challenge in most healthcare settings. Healthcare workers in SA are three times more likely to contract TB compared to the general population [7,19,20,23]. There are other reports of HCWs (3.6%) who contracted TB during employment in a hospital [16]. Three district hospitals in KZN province reported 9% TB cases using employee medical records between 2006 and 2010, with a twofold greater incidence for those working in TB wards [20]. Pulmonary TB was diagnosed in 80% HCWs, of whom 10.0% had multidrug-resistant TB [41]. Thirty-five TB cases were also reported among healthcare workers in PHCs in KwaZulu-Natal province [17].

The infectious dose to acquire TB is not yet known, but it has been estimated in several studies [26,62]. Sornboot et al. [31], reported the estimated exposure time to infectious dose to be high in the sputum room (<15 min) and 24 h in a patient ward, confirming potential risk of HCWs acquiring TB in these areas. Studies reported the use of hospital TB disease or cases as an indicator of TB transmission in the workplaces [1,7,20,63] and focussed on TB case management [5,64]. Health systems employing health workers are responsible for training and protecting them from acquiring TB at work, as well as encouraging reporting of disease [9,45]. TB rate among HWs may be underestimated as some fear stigmatisation [15] and receive care in the private sector and fail to disclose to their employers [7].

The current study demonstrates the importance of early preventive strategies in TB transmission through observations, environmental monitoring, and implementation of a hierarchy of controls. Site assessment and observation of work practices to support the questionnaire data is also necessary, as the discrepancy between the two has been shown in previous assessments [16]. Environmental and engineering control measures must be used together with administrative controls and respiratory protection to mitigate TB transmission in healthcare facilities [9,11,37].

Although airborne MTB was detected and quantified through qPCR, the detection of this pathogen cannot be entirely interpreted as a possible source of infection as qPCR is unable to distinguish between viable and non-viable cells [65,66]. Nonetheless, the environmental detection of this is a public health concern as the viable but non-culturable (VBNC) bacteria that would not be detected using the gold standard culture-based methods are detected through this technique. The VBNC bacteria pose a threat as they can still resuscitate when the environment becomes conducive for active growth and cause disease.

The limitation of our study is that the positive samples were not analysed further for viability and multidrug resistance (MDR-TB). The estimated time for exposure to infectious dose was also not determined. The number of patients and proportion of infectious patients per day (patient flow), as well as workers and visitors on-site during sampling days were not recorded for facilities. The prevalence of TB among health workers was not quantified in our study although the number of TB cases in the last 12 months was recorded per facility. Seasonal variation for airborne TB monitoring was not done in-depth due to the nature of this study. Lastly, the self-administered IPC questionnaires can be biased due to the provision of favourable responses.

## 5. Conclusions

In conclusion, exposure to airborne MTB in healthcare settings is a concern as HWs are at greater risk of contracting TB at work; therefore, aggressive IPC measures coupled with occupational health services are needed to protect them. During pandemics, healthcare facilities (HCFs) can become hotspots and, therefore, control measures need to be regularly evaluated. The current COVID-19 pandemic has also raised the importance of IPC in HCFs for both HWs and the public, and this should be used as the opportunity to rethink IPC for all airborne pathogens. The delay to diagnose and/or misdiagnose, detect, and notify TB during pandemics may further worsen the risk of transmission or spread of TB among HWs.

This study provided evaluated data on existing control measures using questionnaires and compared it with airborne MTB DNA detected using real-time PCR (qPCR) to assess the risk of airborne MTB exposure in healthcare facilities. These findings can be used to inform HCF leadership and staff to improve the administrative, environmental, clinical, and occupational health practices. The study has also confirmed detectable levels of airborne MTB in risk areas despite training on the IPC policy and TB infection control plan reported in all hospitals; therefore, current operational policies and IPC strategic frameworks need thorough review to identify the gaps resulting in poor adherence of IPC measures and to mitigate the risks of exposure to MTB.

## Figures and Tables

**Table 1 ijerph-18-10130-t001:** Elements of tuberculosis infection prevention and control by healthcare facility type.

TB IPC Variable	TB Hospital (N = 4) n (%)	District Hospitals (N = 3) n (%)	Primary Health Clinics (N = 2) n (%)
**Administrative Controls**			
Bed occupancy	925 Hospital B (263) Hospital K (662)	335 Hospital C ward 497 (24) Hospital F (311)	450 Clinic C
Number of health workers	2101	1588 (49 Hospital C ward 497)	734
IPC policy	4 (100)	3 (100)	1 (50.0)
Staff trained on IPC policy	4 (100)	3 (100)	1 (50.0)
TB risk assessment conducted	3 (75.0)	3 (100)	1 (50.0)
Open-window policy available	4 (100)	0 (0.0)	1 (50.0)
Training on TB IPC	4 (100)	3 (100)	2 (100)
Dedicated nurse to treat TB patients	2 (50.0)	2 (66.7)	2 (100)
TB register on site	3 (75.0)	1 (33.3)	0 (0.0)
**Environmental Controls**			
Natural ventilation exists (open windows on opposite walls)	4 (100)	1 (33.3)	2 (100)
Windows always open directly to the outside	4 (100)	1 (33.3)	1 (50.0)
Mechanical ventilation	3 (75.0)	3 (100)	0 (0.0)
Dedicated sputum production area	3 (75.0)	1 (33.3)	2 (100)
Air cleaning method using UVGI was available	3 (75.0)	1 (33.3)	0 (0.0)
Were these UVGI devices functional?	3 (75.0)	1 (33.3)	N/A
HEPA Negative pressure	0 0	1 (33.3) 1 (33.3)	0 0
**Clinical Controls**			
TB screening on arrival	1 (25.0)	2 (66.7)	2 (100)
Designated area for screening	1 (25.0)	2 (66.7)	2 (100)
TB education material for patients	3 (75.0)	1 (33.3)	2 (100)
TB patients isolated in separate room	0 (0.0)	2 (66.7)	2 (100)
**Occupational Health Controls**			
Respiratory protection programme exists	4 (100)	2 (66.7)	1 (50.0)
All HCWs at risk of TB use RPE	3 (100)	3 (100)	2 (100)
N95 respirators available	4 (100)	3 (100)	2 (100)
All HCWs had a respirator fit testing	3 (75.0)	0 (0.0)	0 (0.0)
All staff members screened for TB	4 (100)	2 (66.7)	2 (100)
Baseline chest X-rays done for staff	4 (100.0)	1 (33.3)	1 (50.0)
Occupational health services accessible and available	4 (100.0)	2 (66.7)	2 (100)
Facilities having HWs diagnosed with TB (past 12 months)	3 (75.0)	2 (66.7)	2 (100)
Number of HWs diagnosed with TB (past 12 months)	20	10	7
Incidence rates by facility type *	952	630	954

* Incidence rates (per 100,000 population) was calculated using population rate reported between the period of sampling of March to October 2017 in all facilities.

**Table 2 ijerph-18-10130-t002:** Distribution of study characteristics by presence and absence of airborne *Mycobacterium tuberculosis* (MTB).

Variables	Airborne *Mycobacterium tuberculosis* (MTB), n (%)	
	Not Detected (Negative) 578 (97.8%)	Detected (Positive) 13 (2.2%)
**Province**		
Low-burden TB region (Gauteng)	306 (98.7%)	4 (1.3%)
High-burden TB region (Western Cape and KwaZulu-Natal)	272 (96.8%)	9 (3.2%)
**Healthcare facilities**		
TB hospitals	329 (98.8%)	4 (1.2%)
District hospitals	165 (94.8%)	9 (5.2%)
Primary health clinics	84 (100%)	-
**Seasons**		
Summer	132 (100%)	-
Winter	446 (97.2 %)	13 (2.8%)
**UVGI**		
Absent	308 (97.2%)	9 (2.8%)
Present	270 (98.5%)	4 (1.5%)
**Ventilation type**		
Natural	437 (98.4%)	7 (1.6%)
Mechanical	141 (95.9%)	6 (4.1%)
**Environmental parameters**	**Median (IQR)**	**Median (IQR)**
Temperature (°C)	20.90 (19.40–24.10)	22.10 (21.00–22.70)
Relative humidity (%RH)	46.30 (35.50–55.90)	53.60 (50.00–56.10)
Carbon dioxide (ppm)	619.50 (574.00–697.00)	788.00 (756.00–967.00)
Velocity (m/s)	0.11 (0.06–0.18)	0.07 (0.04–0.14)
Air changes per hour (ACH)	0 (0–0)	0 (0–0.34)

MTB: *Mycobacterium tuberculosis*, UVGI: ultraviolet germicidal irradiation, IQR: interquartile range.

**Table 3 ijerph-18-10130-t003:** Positive MTB samples categorised by province, facility type, and department.

Healthcare Facility	Department	TB (DNA Copies/m^3^)
**Gauteng Province**		
District hospital	TB ward (CH)	3.43 × 10^6^
TB-specialised hospital	TB ward (S)	4.74 × 10^6^
TB-specialised hospital	TB ward (S)	4.79 × 10^6^
TB-specialised hospital	TB ward (S)	7.10 × 10^6^
**Western Cape Province**		
District hospital	Medical ward: females (R2)	1.08 × 10^4^
District hospital	Medical ward: males	1.19 × 10^4^
District hospital	Medical ward: males	6.41 × 10^4^
District hospital	Medical ward: females (1)	6.44 × 10^5^
TB-specialised hospital	Medical ward: females (R2)	9.42 × 10^4^
District hospital	Casualty area (Em)	2.20 × 10^5^
District hospital	Casualty area (Em) (HV2R1)	6.82 × 10^6^
District hospital	Casualty area (Em) (HV2R2)	6.93 × 10^6^
District hospital	Casualty area (Em)	3.55 × 10^7^

**Table 4 ijerph-18-10130-t004:** Comparison of microclimate parameters (medians and interquartile ranges, IQRs) by categorical indicators.

Variables	Temperature (°C)	Humidity (%RH)	Carbon Dioxide (ppm)	Velocity (m/s)
**Province**				
Low-burden TB region ^a^	21.00 (19.50–25.30)	38.20 (30.00–45.00)	605.00 (547.00–766.00)	0.09 (0.05–0.15)
High-burden TB region ^b^	20.90 (19.30–23.00)	55.60 (50.80–63.70)	627.00 (595.00–674.00)	0.12 (0.07–0.21)
*p*-values	0.0992	<0.0001	0.0294	<0.0001
**Healthcare facilities**				
TB hospital	19.60 (18.80–20.30)	50.40 (36.20–60.20)	596.00 (555.00–635.00)	0.12 (0.07–0.19)
District hospital	22.30 (20.80–25.30)	38.90 (32.20–45.60)	726.00 (597.00–803.00)	0.07 (0.04–0.14)
Primary health clinic	26.00 (24.10–26.80)	55.40(45.85–65.30)	674.00 (602.00–1301.00)	0.12 (0.075–0.19)
*p*-values	0.0001	0.0001	0.0001	0.0001
**Seasons**				
Summer	22.50 (20.45–24.25)	58.80 (52.75–64.70)	604.50 (590.00–672.50)	0.12 (0.07–0.21)
Winter	20.70 (19.20–23.50)	43.90 (33.70–50.70)	627.00 (570.00–731.00)	0.10 (0.05–0.17)
*p*-values	<0.0001	<0.0001	0.5132	0.0135
**UVGI**				
Absent	21.50 (19.70–23.90)	50.50 (35.90–58.10)	657.00 (597.00–792.00)	0.09 (0.06–0.16)
Present	19.90 (18.90–24.30)	44.70 (34.90–51.60)	593.50 (550.00–641.00)	0.12 (0.06–0.20)
*p* value	0.0001	0.0002	<0.0001	0.0265
**Ventilation type**				
Natural	20.30 (19.20–22.60)	49.40 (35.50–59.20)	619.00 (577.00–693.50)	0.12 (0.06–0.18)
Mechanical	24.10 (20.80–25.50)	44.80 (36.90–49.80)	634.00 (576.00–762.00)	0.09 (0.05–0.18)
*p* values	<0.0001	0.0018	0.3165	0.1292

^a^ Gauteng; ^b^ Western Cape and KwaZulu-Natal.

**Table 5 ijerph-18-10130-t005:** Factors associated with airborne *M. tuberculosis* exposure in healthcare facilities.

Variables	Odds Ratio	*p*-Values	95% Confidence Intervals	
**Number of days ***	1.52	0.245	0.749	3.098
**UVGI ***				
Absent	1 (ref)			
Present	0.80	0.770	0.182	3.526
**Province**				
Low-burden TB region ^a^	1 (ref)			
High-burden TB region ^b^	4.82	0.038	1.091	21.358
**Ventilation**				
Natural	1 (ref)			
Mechanical	4.77	0.013	1.398	16.280
**Carbon dioxide**	1.003	0.010	1.000	1.005

* Kept in the final model; ^a^ Gauteng; ^b^ Western Cape and KwaZulu-Natal; adjusted for province, ventilation type, temperature, humidity, carbon dioxide, velocity, air change, UVGI, and number of days; interaction term (UVGI and number of days), (carbon dioxide and number of days).

## Data Availability

Data are available upon reasonable request and within the prescripts of the Protection of Personal Information Act (POPIAct).

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
