# Peer review of "Occupational Risk of Airborne Mycobacterium tuberculosis Exposure: A Situational Analysis in a Three-Tier Public Healthcare System in South Africa"

_ijerph, 2021, doi:10.3390/ijerph181910130_

Round 1

Reviewer 1 Report

Revision of the manuscript “ Occupational risk to airborne Mycobacterium tuberculosis exposure: a situational analysis in a three-tier public healthcare 3 system in South Africa”, IJERPH 1375771,

General comments:

This article presents airborne tuberculosis (TB) detection at selected nine public health facilities from  11 three provinces of South Africa. The authors have determined the possible risk factors which contribute to airborne 12 transmissions. The study  has explored  the risk  of airborne MTB exposure in healthcare facilities by evaluating data on existing control measures using questionnaires and comparison with airborne MTB DNA detected using qPCR. Interestingly, the study has also confirmed detectable levels of airborne MTB in risk in spite of training on the IPC policy and TB infection control plan reported in all hospitals. This study will be beneficial as the findings could  be used to review, and identify the gaps to inform management and staff to improve the TB IPC programs.

There are many notes that have to be standardized.

I hope I helped the authors improve the manuscript.

I have few minor comments:

  1. Line 76: A reference for categorizing Gauteng as having a low TB burden could be added.
  2. Line 109: -20 0C shole be -20ᵒC.
  3. Line 119: Italicize M. tuberculosis
  4. Lines 100, 101 have the notation min, and 105 has minute..please standardize
  5. Line 116: In “Goldilux UV – C meter”, reduce the space between UV-C
  6. Line131: Italicize M. tuberculosis
  7. Line143: Remove bracket after M. tuberculosis.
  8. Table 3: Standardize the alignment.
  9. Line 227: “humidity measurements than the other two facilities (P=0.0001)” use lowercase and

    italicized p=0.0001

  1. Table 5 “P values” use lowercase and italicized p=0.0001

Author Response

Point 1 Line 76: A reference for categorizing Gauteng as having a low TB burden could be added. reference added

Point 2 Line 109: -20 0C shole be -20ᵒC. corrected

Point 3 Line 119: Italicize M. tuberculosis corrected

Point 4 Lines 100, 101 have the notation min, and 105 has minute. Please standardize corrected

Point 5 Line 116: In “Goldilux UV – C meter”, reduce the space between UV-C corrected

Point 6 Line131: Italicize M. tuberculosis corrected

Point 7 Line143: Remove bracket after M. tuberculosis. corrected

Point 8 Table 3: Standardize the alignment. corrected

Point 9 Line 227: “humidity measurements than the other two facilities (P=0.0001)” use lowercase and italicized p=0.0001 corrected

Point 10 Table 5 “P values” use lowercase and italicized p=0.0001 corrected

Reviewer 2 Report

General comments:

I am glad to review this report. The authors collected lot of samples for detecting airborne MTB among various kind of facilities. Results, including detection of airborne MTB in casualty area, are interesting, and may be provide useful information for occupational health interventions in South Africa. However, detection of Mycobacterium tuberculosis (MTB) by PCR does not necessarily be identical to high risk of contracting disease of tuberculosis: This should be further discussed.

.

Specific comments:

  1. Please consider revision of expression of “airborne TB” to “airborne MTB” in abstract, conclusions, and other sections.
  2. In line 57, kindly add the original words of IPC.
  3. In Table 1, a figure for “Dedicated nurses to treat TB patients” reads “66.7.0”.
  4. In Table 1, number of HCWs diagnosed are shown. Please add incidence rate by facility type. How about that in casualty area?
  5. In line 264, “District hospitals reported patient screening (100%).” Please explain the relevant figure in Table 1.
  6. In discission section, please note more definitely whether description is about the present study or about the preceding literature.
  7. Please set part of limitation in discussion section, and describe limitations of the present study. Please also discuss about characteristics of PCR, there.
  8. In line 325, the figure “98.41%” is more detailed than that shown Table 1.
  9. Reconsider the expression of “TB risk factors” in conclusion of abstract.

Author Response

Point 1 Please consider revision of expression of “airborne TB” to “airborne MTB” in abstract, conclusions, and other sections. Revised and corrected in the entire document.

Point 2 In line 57, kindly add the original words of IPC. Added full name

Point 3 In Table 1, a figure for “Dedicated nurses to treat TB patients” reads “66.7.0”. Corrected, the last digit deleted.

Point 4 In Table 1, number of HCWs diagnosed are shown. Please add incidence rate by facility type. The number of incidence rates calculated using the number of cases reported in the last 12 months and number of health workers as reported in the IPC questionnaire (Table 1). Please refer to table below, the incidence rates have been added at the end of Table 1. The numbers may not be accurate as for example PHC, only one provided the number of health workers in the questionnaire.

Type of facility

No of TB cases

Population at risk

Incidence rate (per 100 000 population)

TB hospital

20

2101

952

District Hospital

10

1588

630

Primary Health Clinics

7

734

954

*Population at risk is the number of HCWs employed in the hospitals during the year 2017

How about that in casualty area? The IPC questionnaires were administered per facility, therefore the reported cases were for the entire facility and not per department.

Point 5 In line 264, “District hospitals reported patient screening (100%).” Please explain the relevant figure in Table 1. This was a transcription error and has been corrected and rephrased, we appreciate the reviewer for picking it up.

Point 6 Line131: In discussion section, please note more definitely whether description is about the present study or about the preceding literature. We are not sure of what is required for this comment, however have attempted to add “Our results of this study” for our findings where necessary. Each aspect however first start with our findings, followed by the supporting literature.

Point 7 Please set part of limitation in discussion section, and describe limitations of the present study. Please also discuss about characteristics of PCR, there. The limitations were on the submitted draft but we somewhat missing in the draft manuscript. We have added this section under the discussion.

Point 8 In line 325, the figure “98.41%” is more detailed than that shown Table 1. Corrected. we have decided to use only one decimal.

Point 9 Reconsider the expression of “TB risk factors” in conclusion of abstract. Rephrased, not sure if it is what the reviewer was requesting. We were trying to meet the word limit criteria for the abstract.
